# Impact of Diet on Symptoms of the Irritable Bowel Syndrome

**DOI:** 10.3390/nu13020575

**Published:** 2021-02-09

**Authors:** Robin Spiller

**Affiliations:** NIHR Nottingham Biomedical Research Centre, Nottingham NG7 2UH, UK; robin.spiller@nottingham.ac.uk; Tel.: +44-(0)115-8231032

**Keywords:** irritable bowel syndrome, diet, FODMAP, fibre, allergy

## Abstract

Irritable bowel syndrome (IBS), with its key features of abdominal pain and disturbed bowel habit, is thought by both patients and clinicians to be strongly influenced by diet. However, the complexities of diet have made identifying specific food intolerances difficult. Eating disorders can masquerade as IBS and may need specialist treatment. While typical food allergy is readily distinguished from IBS, the mechanisms of gut-specific adverse reactions to food are only just being defined. These may include gut-specific mast cell activation as well as non-specific activation by stressors and certain foods. Visceral hypersensitivity, in some cases mediated by mast cell activation, plays a key part in making otherwise innocuous gut stimuli painful. Rapidly fermented poorly absorbed carbohydrates produce gaseous distension as well as short-chain fatty acids and lowering of colonic pH which may cause symptoms in IBS patients. Limiting intake of these in low FODMAP and related diets has proven popular and apparently successful in many patients. Existing diet, colonic microbiota and their metabolic products may be helpful in predicting who will respond. Wheat intolerance may reflect the fact that wheat is often a major source of dietary FODMAPs. It may also be either a forme fruste of coeliac disease or non-specific immune activation. Wheat exclusion can be successful in some of these patients. More research is needed to fully understand the mechanisms of food intolerances and how to best ameliorate them in a personalised medicine approach to diet in IBS.

## 1. Prevalence and Key Features

Symptoms of the irritable bowel syndrome, including episodic abdominal pain and erratic bowel habit are ubiquitous and common in the general population [1]. The combination of specific symptoms meeting established criteria are less common, with 9% meeting the Rome III criteria and just 4.6% meeting the intentionally more stringent Rome IV criteria [2]. Psychological abnormalities, particularly anxiety, are common in those seeking heath care, as they are in organic gastrointestinal diseases, with increasing prevalence as the severity of GI symptoms rise [3]. A large survey of IBS patients in Sweden showed 45% with anxiety and 26% with depression scores exceeding the upper limit of normal [4]. Patients with psychological distress were more likely to have visceral hypersensitivity and fatigue. Combining data from five different specialist centres around the world showed that visceral hypersensitivity as assessed by pain induced by rectal distension was strongly correlated with gastrointestinal symptom severity [5]. 

Since pain is required to meet Rome IV criteria [6], this tends to emphasise psychological factors, since these determine central pain processing, which strongly influence reported pain. Subgroups have been defined by different bowel patterns, with IBS with diarrhoea (IBS-D), IBS with constipation (IBS-C) and IBS with mixed bowel pattern (IBS-M) accounting for most. While these subdivisions are very relevant when considering the response to both foods and drugs which alter bowel habit, an alternative classification which also includes comorbidities is a useful guide to overall treatment [7].

## 2. Patterns of Pain and Link to Food

A characteristic feature of IBS is the variability in bowel pattern and pain. Patients report more variation in stool consistency and less predictability of their stool form from day to day [8]. Daily diaries show that patients can be divided into the four distinct subtypes, namely IBS-D, IBS-C and IBS-M and rarely unclassifiable IBS (IBS-U) with mostly normal stools. However, patients tend to recall more abnormal bowel movements than actually are recorded and surprisingly up to 40% of stools are normal [9]. Symptoms typically occur in episodes and in a 3 month study of 185 IBS patients, average episode durations were 2.1 days for diarrhoea, 4.5 days for constipation, 3.1 days for pain, and 3.5 days for bloating [10]. Interestingly, only 41.6% of constipation episode days and 67.0% of diarrhoea episode days were pain episode days. Thus, while pain is related to bowel habit, it is not always due to abnormal motility. The link between pain and diarrhoea is the easiest to explain since colonic contractions of increasing amplitude often precede bowel movements [11]. Colonic contractions have been linked temporally with pain supporting the idea that at least some pain is a response to increased bowel wall tension due to contractions, combined with visceral hypersensitivity [12]. 

Ambulatory 24 h recordings make it clear that eating is a major stimulus to colonic contraction [13], which becomes quiescent during sleep [14]. This is in keeping with the observation that symptoms rarely wake patients from sleep and are often provoked by eating. A previous study, in which patients recorded symptoms, eating and bowel movements during 7 days, suggested that pain was more clearly linked to eating than defecation [15]. Fasting is often used to stop a severe flare in the condition and many patients adopt unusual eating patterns to avoid symptoms interrupting their planned activities. Many skip midday meals to avoid having symptoms at work though this has rarely been systematically studied. One large survey in Japan reported that IBS patients were more likely to skip breakfast and/or lunch compared to healthy controls [16]. 

The reasons for pain in constipation is less clear though recent surveys have suggested that pain is more of a problem in IBS-C than other subtypes [17]. The pattern of pain may be different for different IBS subtypes. While 83% of IBS-D symptom flares occurred within 15 min of a bowel movement, this was only true of 36% of IBS-C episodes [18]. Pain in IBS-C progressively increases with number of days since last bowel movement [9], suggesting that colonic distension or the motility patterns associated with distension may be important. 

## 3. Immune and Non-Immune Mechanisms Underlying Postprandial Symptoms

There are multiple possible mechanisms whereby eating could stimulate pain (Figure 1). These different mechanisms produce a different pattern of symptoms and different time courses, ranging from virtually immediate (0–30 min) to days later. This variable time lag makes it particularly difficult for either patient or clinician to identify culprit food items. The simplest mechanism, but perhaps the least common in a GI clinic, is food allergy typically caused by shell fish, peanuts, tree nuts (walnut, almond, hazelnut, pecan, cashew and pistachio) and fish [19]. This is based on specific IgE antibodies bound to the surface of mast cells found throughout the body and easily detected by a simple blood test. Orally ingested antigen activates mast cells throughout the body, giving rise to urticaria, asthma as well as gut symptoms, usually pain and diarrhoea. 

However, mast cell activation can be limited to the gut, where their activation is more difficult to document and traditional allergy tests such as specific serum IgE or skin prick tests are negative. Recent studies show that after infectious gastroenteritis, mast cells in the murine gut can react to food given at the time of infection [20]. Importantly, specific IgE was not found in serum, which would mean that current tests used to detect specific allergy would be negative. These gut mast cells can also be activated by psychological stressors [21] as well as directly by lectins such as raw potatoes [22] and agglutinins from uncooked red kidney beans [23], though these laboratory studies have yet to be shown to be clinically relevant by properly controlled trials of dietary intervention. At the non-immune end of the spectrum, there is food intolerance due to FODMAPs, where gaseous distension and/or other metabolic products such as lactate or pathogen-associated molecules such as LPS may cause symptoms, particularly in viscerally hypersensitive IBS patients. 

The time course of symptoms will vary depending on the mechanisms. At least some food-associated pain could be due to indirect stimulation of colonic motility by the act of eating. This would occur within 5–30 min, while effects due to colonic distension would be predicted to develop after 4–6 h, as food residue enters the colon. Onset between 30 min and 4 h might suggest a direct effect of food on the small bowel as demonstrated by recent studies using confocal laser endomicroscopy [24]. Direct instillation of foods into the duodenum during confocal laser endoscopy (CLE) showed that certain foods could induce acute increases in lymphocytes and permeability with increases in the tight junction protein, claudin 2, typically associated with leaky epithelia [24]. The IBS patients in this study were selected because they felt certain foods triggered symptoms. Of those that responded to food challenge, a larger proportion (69%) had a personal and/or family history of atopy than those that did not respond (38%). The commonest food inducing a permeability response was wheat, which accounted for 60% of responses, while yeast (20%), milk (9%), and soy (7%) were less common [24]. The authors reported a good response to an exclusion diet but this was open label and for the most part a wheat exclusion diet. This is often successful so whether the response is specific to CLE-positive patients is uncertain. It should be noted that although these patients lacked evidence of systemic allergy in the form of serum IgE antibodies, it remains possible that there is localised allergy restricted to the gut mucosa [20]. Activation of mast cells locally could both increase gut permeability and also, through release of their numerous mediators, particularly histamine and prostaglandins, activate enteric nerves to generate pain. 

Impaired barrier function has been associated with both visceral hypersensitivity and pain severity in IBS patient with diarrhoea (IBS-D) though the precise mechanism is unclear [25]. The same group showed, in a group of post-infectious IBS with impaired barrier function, that glutamine treatment improved both barrier function and symptoms [26]. A more recent study using a combination of xyloglucan and pea protein and grape tannins supports the idea that enhancing barrier function might be therapeutic in IBS-D patients [27].

## 4. Role of Mast Cells in IBS

Impaired barrier function has also been noted as a stress response, which has been shown in humans to be mediated by activation of mast cells [21]. These play a key role in atopy which is more common in IBS than controls without any functional gastrointestinal disease, with an odds ratio mean (95% confidence interval) of 1.4 (1.3–1.6) [28]. Detailed mechanistic studies of caecal biopsies from IBS-D patients with atopy show that compared to non-atopic IBS-D patients, they have more severe symptoms with higher numbers of mucosal mast cells and greater release of mast cell tryptase from incubated biopsies and greater mucosal permeability [29]. Mast cell numbers correlated both with biopsy permeability and symptom severity, suggesting that mast cell mediators including histamine, tryptase and prostaglandins could be causing symptoms. 

Whether mast cell changes in the small bowel are also reflected in mast cells in the colon is unclear but an increase in colonic mast cells is one of the most consistent changes found in IBS [30] which has also been noted in a few studies of duodenal biopsies [31,32]. Furthermore, the number of mast cells in close proximity to enteric nerves has been correlated to the severity of pain in IBS [33]. Studies of mucosal mediators released from colonic biopsies have shown that they activate afferent enteric nerves [34]. The nerve response correlates with pain severity, though no single mediator appears dominant with both cytokines and mast cell products possible (Lam, in press Clinical Translational Gastroenterology 2021).

## 5. Impact of Colonic Fermentation on Gut Symptoms

FODMAPs, being poorly absorbed, enter the colon, where they are rapidly fermented. This can be readily visualised by MRI, which shows how the osmotically active fructose distends the small bowel with fluid and subsequently the colon, where it produces gas [35] along with a rise in breath hydrogen [36]. Larger fructan polymers such as inulin have little impact in the small bowel but rapidly increase colonic gas, distending the ascending colon and increasing symptoms of flatulence and the sensation of distension in IBS patients [37]. These acute studies using large doses of fructans confirm longer-term clinical trials with smaller doses, which show increased flatulence and gas [38,39]. Fermentation of fructose polymers with a range of degree of polymerisation (DP) in *in vitro* models using human faecal inoculates shows that speed of fermentation, total gas production and SCFA production slows as the DP increases [40]. Rapid fermentation is associated with a fall in pH to as low as 4.5, due to accumulation of lactate faster than it can be metabolised. Animal studies, albeit often using excessive doses unlikely to be feasible in humans, show that high FODMAP diets are pro-inflammatory, aggravating colitis [41] and inducing visceral hypersensitivity [42]. Low FODMAP diets reduce urinary histamine while high FODMAP diets increase urinary histamine [43], suggesting that FODMAPs’ metabolic products can cause mast cell activation. This could be mediated by short-chain fatty acids [44] or indirectly via changes in the microbiota. Slowing transit using viscous fibres can reduce gas production [45] and this may be a future way of improving tolerance of FODMAPs. 

## 6. Clinical Approach

One of the first things a clinician needs to do with a patient who has been diagnosed with IBS is to assess their eating patterns and diet. Several small food frequency surveys in Sweden have indicated that IBS patients do adapt their diet according to their beliefs about what causes symptoms, which may result in a nutritionally deficient diet, particular due to reductions in dairy products [46,47]. The same surveys found that the most commonly avoided foods were milk, onions and cabbage along with many other idiosyncratic items, though these were not related to objective evidence of specific allergy nor lactose malabsorption [47]. Plainly identifying specific food intolerances is difficult because of the complexity of food intake and the fact that foods, such as milk for example, may have different effects if taken alone on an empty stomach of if mixed with a large meal, when the rate of delivery of lactose will be much reduced [48]. Furthermore, adverse reactions to food such as excessive gas may be delayed [49] making it difficult for the patient to identify the food responsible. As well as taking a history of any perceived intolerances, it is always worth a brief enquiry about caffeine intake which can be extremely high. Caffeine has been shown by colonic manometry to stimulate colonic motility [50,51] and this may well be propulsive in effect since it has been shown to shorten post-operative ileus [52]. Most clinicians will have seen cases where diarrhoea is apparently cured by reducing intake, so this simple manoeuvre is well worth a brief trial if intake appears to be excessive. 

## 7. Eating Disorders

Eating pattern and beliefs about weight and self-image should be explored in all patients since there is a danger of patients with eating disorders accumulating an ever increasing list of foods which are excluded. If this is suspected, then asking the patient to keep a daily food diary to bring to the next appointment can be very helpful by demonstrating a very restricted diet. Patients whose problem is primarily occult anorexia and/or bulimia are highly likely to have either functional constipation, found in 24% or IBS, mostly constipation or mixed subtype, reported in 46% [53]. The opposite condition of binge eating is associated with diarrhoea [54]. It is also worth enquiring specifically about the fat content of the diet since fat stimulates bile and pancreatic secretions and increases small bowel water [55] and also is a potent stimulator of the colonic response to feeding [13]. This may in part explain why high fat intake was found to increase the risk of diarrhoea in a group of obese subjects [54]. IBS patients with diarrhoea tend to be overweight, which is associated with faster whole gut transit and increased urgency and stool frequency though whether this is related to high fat intake is unknown [56].

If eating patterns are very disturbed and seem to be causing the underlying bowel disturbance, their management will require referral to specialist clinics. Assuming primary eating disorder has been excluded, the next step will either be a trial of drugs such as antispasmodics/low-dose tricyclic antidepressants/soluble fibre or dietary intervention. This article will not consider medication but focus solely on diet. However, it is worth considering the role of prescribed fibre supplements which are often easier to implement than changing the diet.

## 8. Fibre Supplements

Decreased fibre intake is associated with an increased risk of constipation in the general population [57] and several randomised trials suggest that there is benefit to some IBS patients by giving fibre. However, two meta-analyses agree that this effect is only seen in trials of viscous fibre and particulate fibre such as wheat bran shows no benefit [58,59]. Recent studies show how viscous fibre (psyllium) acts by trapping water in the small bowel and increasing colonic water content, leading to softer more frequent stools [60]. Why viscous fibre would benefit constipation is obvious but why it also benefits non-constipated patients is unclear unless viscous fibre slows digestion and reduces rapid fermentation. Wheat bran was in general less well tolerated than psyllium and more likely to result in drop out because of side effects [61], particularly flatulence [62]. Once simple modifications to the diet such as ensuring adequate fibre intake and avoiding excessive caffeine have been tried, the next step is referral to a dietician for a full dietary history and consideration of an exclusion diet.

## 9. Exclusion Diets: Principles and Practice

### Elimination Diets

The idea that specific foods exacerbate IBS symptoms forms the basis of the elimination diets pioneered by the Cambridge group in the 1990s which focused on excluding poorly absorbed starches with the aim of reducing colonic gas production [63]. This approach was further developed into a more empirical approach in which patients started by eliminating foods commonly observed to cause symptoms. These included dairy products, cereals, citrus fruits, potatoes, tea, coffee, alcohol, additives and preservatives. The diets were mostly based on fresh meat, fish, rice vegetables and goat, sheep, or soy milk. If symptoms remitted on the exclusion diet then foods were re-introduced, a new food group every 2–3 days and in this way foods were classified as being tolerated or as causing symptoms. Symptoms remitted on the elimination diet in 91/189 (48%) of patients and, after subsequent re-challenge, 73 were able to identify one or more food intolerances and 72 remained well on follow up of more than 1 year [64]. While this proved the principle, implementation was difficult because there were no simple rules. Each diet was entirely individual and empirical and required considerable dietician input, which was impracticable for most clinicians. These studies however provide a list of commonly identified food intolerances (Table 1). More recent surveys of patients perceptions of which foods caused symptoms identified similar overlapping lists which vary by country, no doubt related to the differing diets. One Swedish survey found that 84% of IBS patients reported at least one food that triggered their symptoms. These included dairy products (49%), beans/lentils (36%), apple (28%), flour (24%) and plum (23%) together with foods rich in biogenic amines—wine/beer (31%), salami (22%) and cheese (20%)—with 52%reporting fried/fatty foods in general [65]. A population-based survey in Norway reported that on average IBS patients avoided 2.5 food items. This included 35% who avoided milk, 14% cheese, 16% pulses, 24% onions, 10% wheat flour, 26% coffee and 12% beer [47]. There was no association between the number of foods identified and mood disorders nor musculoskeletal pain. Objective tests were poor at identifying intolerances, thus only 6.5% of those who reported milk intolerance had lactose malabsorption, as shown by a positive lactose breath hydrogen test, and none with reported egg intolerance had IgA antibodies to egg albumin, while only 19% who reported gluten intolerance had gliaden antibodies [47].

## 10. Exclusion Diets Based on Immunological Tests

Initial studies focused on the idea that IBS patients had mast cell-driven food allergy. Indeed, as a group, IBS patients do have a slightly increased incidence of atopy compared to non-IBS patients. A large primary care study in the UK reported that 44.8% of IBS had atopy compared to 32.7% of healthy controls, the strongest effect being seen for allergic conjunctivitis and hay fever (odd ratio [95% CI 2.98 [2.6–3.37]) [28]. Typical mast cell-driven systemic food allergy characterised by rash, urticarial and other allergic phenomena up to anaphylactic shock is easily recognised and such patients usually attend allergy clinics rather than gastroenterological ones. A large survey of gastroenterological outpatients found that while 32% complained of adverse reactions to food, and 14% had suggestive criteria such as atopy, eosinophilia and elevated IgE to specific food antigens, only 3% had a confirmation of allergy by endoscopic allergen provocation and/or response to dietary challenge [66]. 

While attempts to show evidence of specific systemic food allergy using serum antigen-specific IgE antibodies and skin prick tests have largely failed to show differences between IBS and controls, it remains possible that gut-specific allergy is responsible for symptoms which follow soon after food ingestion. Recent animal studies suggest that exposure to ovalbumin during a bout of gastroenteritis can lead to a specific ovalbumin allergy limited to the gut with no IgE antibodies detectable in the serum [20]. This may be relevant for those IBS patients who react to instilled wheat flour and/or milk or soy protein with increased gut permeability, despite lacking specific IgE and/or positive skin prick tests [24]. Alternative routes to activation of mast cells other than by binding of antigens by mast cell surface bound IgE include psychological stress and a non-specific immune activation via TLR4 activation, as has been reported for wheat amylase/tryptase inhibitors [67]. 

A search for simple blood indicators of food intolerance lead to a number of studies using food-specific IgG levels. Despite initial enthusiasm based on small clinic populations suggesting abnormal levels of IgG4 antibodies to food [68] subsequent larger population-based studies of IBS patients with adequate controls found no difference. The varying titres appeared to reflect dietary intake [69]. The apparent success of a diet eliminating food based on the presence of specific IgG4 antibodies [70] may have been related to the fact that 84% of the diets excluded milk and 49% excluded wheat. Both these interventions have considerable success regardless of immunoglobulin levels since this relies on removal of poorly absorbed carbohydrate (lactose and fructans, respectively which cause symptoms by gas and distension) rather than excluding specific allergens. 

## 11. Lactose and Fructose Restriction

The role of lactose in causing symptoms in lactase-deficient individuals was clearly established in the 1970s with the demonstration that lactose caused an osmotically driven influx of fluid into the small bowel [71] and subsequent acceleration of small bowel transit, delivering undigested lactose to the colon, where it was rapidly fermented. Clinical studies showed that faster orocecal transit was associated with worse symptoms. This suggests that the rate of delivery of fermentable carbohydrate to the colon as well as the severity of malabsorption are important in determining symptoms [72]. Serial studies suggest that some adaptation to continued lactose ingestion occurs with a decrease in breath hydrogen response and a decrease in symptoms after 3 weeks of exposure to lactose in the diet [73]. This is likely to represent adaptation of the microbiota. Recent fecal incubation studies also indicate that those that are symptomatic produce more lactate and other short-chain fatty acids, suggesting again that the colonic microbiota are important determinants as to whether symptoms develop or not [74]. The other key factor is visceral hypersensitivity to distension and bacterial metabolites [48]. A study in China, where most have genetically determined lactose malabsorption, shows that those with lactose intolerance have similar degrees of breath hydrogen response to lactose as those with asymptomatic lactose malabsorption but differ in reporting symptoms at lower doses [75]. When visceral hypersensitivity was assessed by rectal distension after a lactose challenge, lactose-intolerant patients showed lower thresholds for discomfort [76]. Double-blind, placebo-controlled studies show that even in documented lactose malabsorbers who report being intolerant of milk, quite large amounts of lactose, equivalent to 240 mL of milk, can be tolerated with no difference from a lactose-free placebo [77]. Furthermore, many patients take only small amounts of lactose, which may in part explain why the response to lactose exclusion correlates poorly with evidence of lactose malabsorption [78]. IBS patients with lactose intolerance often also absorbed fructose poorly and this could cause symptoms in some as could sorbitol [79]. Fructose and sorbitol reduced diets showed benefits in uncontrolled interventions [80,81] leading the way to a more comprehensive diet which reduces all sources of poorly digested, rapidly fermentable carbohydrates. Milk intolerance is more common than just lactose malabsorption and in children there is clear evidence of a true allergy to cow’s milk protein which is reduced by hydrolysing the casein component [82]. Classical IgE mediated milk protein allergy is rare in adults but recent trials have suggested that hydrolysing milk protein can reduce flatulence in patients with functional bowel disorders [83,84]. It is unclear whether this is due to gut-specific allergy but this has been recently demonstrated in mice for egg albumen [20] and specifically for milk protein in coeliac patients, who responded to rectal instillation of milk protein despite lacking elevated specific serum IgE [85] 

## 12. NICE Diet

This is based on a diet recommended by the National Institute for Health and Care Excellence (NICE) and the British Dietetic Association [86]. While this diet does recommend avoiding certain common foods often associated with symptoms such as spicy and fatty foods, alcohol, coffee, onions, cabbage and beans, it also focuses on eating style. Thus, it encourages regular meals, three times daily, along with 3 snacks and an emphasis on taking time to eat and doing so in a relaxed fashion rather than having hurried irregular meals that patients often report [16]. It also specifically advises against carbonated drinks and foods containing polyol sweeteners such as mannitol, as found in chewing gum, foods also prohibited in the low FODMAP diet. It is worth noting that these measures are based on consensus rather than evidence since to test all possible variations in a randomised controlled way would require prohibitive numbers to achieve adequate statistical power. Two recent studies [87,88] included a modified NICE diet as a comparator against the low FODMAP diet and found that both diets improved symptoms equally, though of course without a control arm, it is not possible to exclude the possibility that the response was largely placebo. Diets empower patients and make them feel more in control, which may alleviate anxiety and improve symptoms in a non-specific way, which is probably why they are so popular with patients.

## 13. Low FODMAP Diet

FODMAP is an acronym for Fermentable, Oligo-Di- and Mono-saccharides and Polyhydric Alcohols, a term which neatly describes foods which often lead to symptoms of discomfort, flatulence and erratic bowel habits. This unifying concept covers both lactose and fructose as well as fructans and other poorly absorbed carbohydrates whose exclusion in the low FODMAP diet has gained tremendous popularity with both dieticians and patients world-wide [89]. 

Dietary interventions are difficult to double blind and prone to powerful placebo/nocebo effects so it was not until the Melbourne group performed a double-blinded randomised placebo-controlled trial using clear solutions of test substances that it was conclusively proven that both fructose and fructans induce more symptoms including abdominal pain, bloating and flatulence in IBS patients than glucose alone [90]. However, it should be noted that the commonest doses taken by patients in this trial were 14 g of fructans and 28 g of fructose. These are substantially higher than a normal diet and were delivered as liquid drinks, which, by speeding gastric emptying, may make symptoms more severe. A recent diet intervention trial in the UK reported that a standard diet would provide approximately 15 g fructose and 5 g fructans [91] so one would predict the effect of interventions to be much less than observed in the initial trial. Nevertheless, a subsequent carefully double-blinded and well-controlled cross-over study compared a low FODMAP diet (3 g/day) with a typical Australian diet containing 23.7 g of FODMAPs per day showed a clear worsening of symptoms on the high FODMAP diet (see Figure 2) [92].

Cross-over studies tend to show bigger differences than parallel-group designs because when subjects swap diets, changes are easier to detect and blinding may be broken. However, meta-analyses of a subsequent 7 RCTs including 397 IBS patients suggest a significant improvement with a low FODMAP diet when compared to both usual diet and a high FODMAP diet, the relative risk (95% CI) of not improving being 0.69 (0.54 to 0.88) [93]. However, the benefit is modest and the increased cost and inconvenience may explain why on average only 41% of patients still adhere to the diet after 6 weeks follow up [94]. Interestingly, adherence was greater in those with IBS-D compared to IBS-C, being 51% versus 10%. This is in keeping with other observations that IBS-D may do slightly better [88] with improved stool consistency in at least one trial [92] though this has not be formally tested in a controlled trial. More worrying is the observation that those that do adhere have a high prevalence of eating disorders, emphasising the importance of dietician supervision to avoid under nutrition [94]. It is also important to recognise that patients should not remain on the strict exclusion diet but once symptoms have responded they should reintroduce suspect foods for a challenge period and thus enable a liberalising of diet. Few studies have dealt with this phase of the treatment but a recent report showed that approximately 50% of patients were successful in achieving symptom relief. This study showed the importance of a dietician because many of the initial perceived intolerances were not confirmed during systematic reintroduction [95].

## 14. Predicting Response to Low FODMAP Diet

Habitual intake is important since those who take very few FODMAPs, for example those already on a gluten-free, low-lactose diet may experience little benefit as they are already excluding most FODMAPs. Given that a low FODMAP diet is thought to reduce flatulence and bloating by inhibiting fermentation of FODMAPs, it is perhaps expected that the initial microbiome could influence response. One short-term feeding trial in children assessed the symptom response to a 2 day intervention of either a low FODMAP (maximum 9 g of FODMAPs/day) or TACD (Typical American Childhood Diet) (maximum 50 g FODMAPs/day) with a 5 day washout period between diets. The diet significantly reduced the number of pain episodes per day. Responders were defined as those achieving a 50% reduction in daily pain episodes on the low FODMAP diet but not the TACD. At baseline, responders compared to non-responders were enriched in the *Bacteroides* genus, *Ruminococcaceae* family, and *Faecalibacterium prausnitzii* species, all known for saccharolytic metabolic capacity [96]. Functional metagenomic predictions made by linking taxonomic information from the 16S rRNA gene sequences to the Kyoto Encyclopedia of Genes and Genomes (KEGG) suggested that responders had more alpha-N-arabinosfuranosidase. This would allow more rapid metabolism of wheat fibre, which might increase symptoms. A small Swedish study comparing a NICE IBS diet (34 patients) to a low FODMAP diet (33 patients) showed that while there were no discernible microbiota differences between responders and non-responders to the NICE diet, responders to the low FODMAP diet did differ in a multivariate analysis using 54 DNA probes targeting >300 bacteria at variable taxonomic levels. Non-responders had greater abundance of certain bacteria both before and after the diet. However, given the small numbers and known huge variability in microbiota, it is unlikely that these specific differences will be reproducible since it is more the metabolic function than the precise taxa that is likely to be impacted by diet. The most obvious correlation between diet and specific bacteria was with Bifidobacterum, which correlated with lactose intake, an important component of the Swedish diet which is significantly reduced by a low FODMAP diet [97]. 

An alternative way of assessing the metabolic capability of faecal microbiota is by analysing the volatile organic compounds (VOC) in the head space of a bottle of stool heated to 50 °C for 10 min. This was used in a trial of a low FODMAP diet versus a sham diet in which response was defined as a fall of >50 on the BS Symptom Scoring System (IBS-SSS). The VOC profile at baseline was shown to be able to predict response rate using levels of 15 compounds that in principle components analysis could account for 25% in the variation in response to a low FODMAP diet [98]. However, linking VOC to specific microbiota is not possible at present so it is hard to know the underlying mechanism of benefit. More useful in the clinical setting is the demonstration that chronic diarrhoea and higher peak breath methane concentrations were predictive of response to the LFD in a group of patients who had a positive lactose or fructose breath hydrogen test [99].

Just recently the exciting potential of genetic testing has been reported, with a reduced efficacy of low FODMAPs diet in patients with IBS-D carrying sucrase-isomaltase (SI) hypomorphic variants (Figure 3). These variants were shown in a gene dose-dependent fashion to reduced response rate to both low FODMAP and NICE diets, which fell from 42.1% in patients with one or no copies to 16.7% in those with two copies of the variant genes [100]. This is in keeping with the idea that symptoms are being driven by malabsorbed sucrose or maltose and these are not excluded in the low FODMAP diet. Future studies using genetic information to predict intolerances including other digestive enzyme mutations would seem indicated. 

## 15. Impact on Microbiome

FODMAPs play an important role in providing nutrients for colonic microbiota which degrade them to short-chain fatty acids. Restricting intake as part of a low FODMAP diet reduces total bacterial numbers, particularly the relative abundance of the Clostridia cluster IVa within the family Lachnospiraceae as well as the species *Akkermansia mucinophilia.* There are also reductions in absolute numbers of Bifidobacteria with increases in *Ruminoccocus torques*. Clostridia cluster IVa are important producers of butyrate [101] which is known to be an important fuel for colonocyte health. Feeding oligofructose and inulin stimulates growth of Bifidobacteria which are important in breaking down such non-starch polysaccharides yielding acetate. This is utilised by other bacteria to produce proprionate and butyrate [102,103]. Bifidobacteria have a number of health benefits [104] including competitive exclusion of pathogen, immune modulation and provision of nutrients from otherwise indigestible dietary components along with improvement in barrier function [105]. They have been used as probiotics, with one trial suggesting benefit in the treatment of IBS [106], so reducing their numbers may be undesirable. 

One concern about restriction of FODMAPs is that by depriving colonic bacteria of carbohydrate substrate, this will lead to a switch from saccharolytic to proteolytic metabolism, increasing the degradation of colonic mucin, particularly in the distal colon, with possible carcinogenic implications [89]. These changes have raised concerns about the long-term risk–benefit ratio of low FODMAP diets [89]. However, high FODMAP diets may also have adverse effects as seen when feeding oligofructose supplements which produce increased faecal lactate and mucosal irritation as evidenced by increased mucin excretion [107]. Most such mechanistic studies have been performed during short-term very substantial dietary interventions. These may overestimate possible harms, since in clinical practice the initial restrictive phase is followed by a re-introduction phase in which many excluded items are allowed back into the diet [108].

## 16. Wheat and Gluten-Free Diets

Wheat has consistently been identified as aggravating IBS symptoms, both as reported in patient surveys [65] and trials of food exclusion [64]. A recent survey in Australia reported that 14.9% of the general population reported wheat intolerance. However, only 1.2% had coeliac disease, suggesting that 92% of those with wheat intolerance do not have coeliac disease. Patients with IBS were 3.5× more likely than controls to have intolerance [109]. Virtually identical figures were obtained in a previous UK survey [110]. The mechanism is unclear but immune reaction to gluten as seen in coeliac patients is associated with IBS symptoms [111] and key abnormalities seen in IBS, such as increased gut permeability and increased mucosal mast cells and decreased serotonin transporter (SERT), are also seen in coeliac patients prior to starting a gluten-free diet [31]. These observations have led many IBS patients to adopt the more demanding and expensive gluten-free diet, whose sales have recently increased substantially. Most guidelines on managing IBS stimulate that it is important to exclude coeliac disease since in unselected IBS patients meeting Rome III criteria there is a 7-fold increased risk of having coeliac disease compared to controls [112]. Coeliac disease is due to a specific immune reaction to gluten, which is a product of wheat protein digestion which binds to HLA DQ2 and 8 receptors on antigen presenting cells activating them to produce a mucosal inflammatory response characterised by lymphocyte infiltration, crypt hypertrophy and villous atropy associated with rapid epithelial cell turnover. This can be diagnosed with 93% sensitivity and 97% specificity by means of IgA endomysium antibodies [113] often confirmed by duodenal biopsy before instituting a life-long gluten-free diet. 

## 17. Non-Coeliac Wheat Sensitivity

After excluding coeliac disease by serology, HLA typing and if necessary duodenal biopsy in patients reporting wheat intolerance, then one is left with non-coeliac wheat sensitivity [114]. Suspicion that some IBS-D patients might be a form fruste of coeliac disease was raised initially by finding that 1/3 IBS-D patients were HLA DQ2 or 8 and 23% had increased intraepithelial lymphocyte counts [115]. A subsequent uncontrolled trial of a gluten-free diet (GFD) was encouraging with a 60% response rate in HLA DQ 2-positive patients compared to 12% in HLA DQ2-negative patients [116]. A randomised, placebo-controlled trial of gluten-free versus gluten-containing diet showed that GFD was associated with reduced bowel frequency, which was significantly greater in those who were HLA DQ2 or 8 positive [117]. Two further trials performed in Australia by the same institution but using slightly different design suggested initially that gluten might cause symptoms [118] and then that this was mostly likely due to a nocebo effect [119]. The same group subsequently performed a challenge study in a different institution comparing gluten versus fructan versus placebo in IBS patients following a GFD in the belief that they were gluten sensitive. This showed that while fructans induced symptoms, gluten did not. This suggests that the benefit of a wheat-free diet is most likely due to excluding fructans rather than gluten [120].

Since the nocebo effect of taking food one believes one is allergic to is substantial [119], such patients can only be objectively identified by the laborious and rarely performed double-blind challenge of encapsulated food. When this is performed, requiring a minimum of 2 days symptoms following exposure, 276 out of 920 (30%), patients with IBS were diagnosed as wheat sensitive [114]. It should be noted that this was in a group who had been selected to undertaken a double-blind challenge presumably on the basis of a clinical indication that food allergy might be relevant and so may overestimate the true prevalence in the general IBS population. As Figure 4 shows, wheat reproduced many of the key IBS symptoms. Of these 270, 70 had wheat sensitivity alone (Group 1) but the remaining 206 (Group 2, labelled multiple food allergic) were also responsive to double-blind challenge with cow’s milk protein and reported intolerance of other foods, most commonly eggs and tomato [114]. Group 2 had features more typical of allergy patients, with a higher frequency of family and personal history of atopy and more eosinophils in the colonic lamina propria. In contrast, Group 1 had some features of celiac disease including anaemia, weight loss, increased duodenal biopsy lymphocytes, HLA DQ2 or 8 haplotype and antigliadin antibodies without the diagnostic duodenal villous atrophy, raising the possibility that full blown celiac might develop in the future. This group may overlap with those studied by Fritscher-Ravens and colleagues [24] and would be predicted to respond well to a wheat-free diet.

The practical implications from these studies are that wheat exclusion may well be beneficial in up to 1/3 of IBS patients but that patients should be initially encouraged to simply minimise their intake of wheat rather than adopt a gluten-free diet, which is expensive and very restricting on social activities. Only if this fails are more elaborate immunological testing required, particularly if there is a family or personal history of atopy. 

## 18. Low-Histamine Diets

While histamine elicits a number of serious adverse effects including most prominently headache, rash, urticarial, diarrhoea and hypotension [121], it seems unlikely to be often confused with IBS. Although some foods such as wine, cheese and fermented vegetables contain significant amounts of histamine, this is rapidly degraded by diamino oxidase enzymes present in the gut so that ingestion rarely elevates plasma histamine [122]. Reproducibility of the response to oral histamine challenge is poor even in individuals who report histamine intolerance. However most such patients see immunologists rather than gastroenterologists [123]. As recent guidelines indicate, despite much inaccurate comment in social media, there are no objective measures to define histamine intolerance. Nor can the degradation pathways via diamino-oxidase or histamine-N methyl transferase be adequately measured at present. Furthermore, there are no randomised trials of either diet nor pharmacological treatment so much work remains to be carried out before any firm recommendations can be made [124]. 

## 19. Laxative Effects of Fruit and Vegetables

Recent imaging studies have shown that lettuce and rhubarb both markedly increase water content of the small bowel and the ascending colon [125]. This secretory response is likely to represent a response to specific compounds such as lactucins and rhein, produced by plants to inhibit grazing by herbivores. Those with IBS-D might benefit from avoiding these as do ileostomists [126]. In contrast, those with constipation might benefit from such effects, which are also seen with whole fruit such as apples [127] and kiwifruit [128]. This mechanistic evidence for kiwifruit is also supported by several randomised placebo-controlled trials showing that the increase in stool water is associated with increased bowel frequency and softening of stools [129,130,131].

## 20. Conclusion and Future Perspectives 

It is clear that taking a careful dietary history including not just what is eaten or avoided but also the meal pattern is important in managing IBS. Patient’s beliefs about what causes their symptoms must be explored since they may be correct, but equally they may need challenging if an eating disorder has developed. Once coeliac and other IBS-mimicking disorders such as inflammatory bowel disease and bile salt malabsorption have been excluded, dietary management should most economically start with the NICE guidelines. This will result in improvement in approximately 40%. If this fails, then there are many possible options. My personal approach is to start with a trial of a simple exclusion diet excluding just a few common items such as wheat, milk, excessive caffeine and/or specific foods that the patient has identified, which is a relatively easy first step. If this fails, then referral to a dietician for a trial of a low FODMAP diet is recommended, with particular emphasis on the need to complete the reintroduction phase of the diet so that the long-term diet is adequate nutritionally. This stepwise approach is both economical and will result in improvement in symptoms in a substantial proportion of IBS patients. Future research should focus on developing patient-acceptable, non-invasive ways of reliably identifying specific food intolerances, particularly wheat and milk protein intolerance, together with ways of reducing both immune- and non-immune-based adverse responses to food in IBS. 

## Figures and Tables

**Figure 1 nutrients-13-00575-f001:**
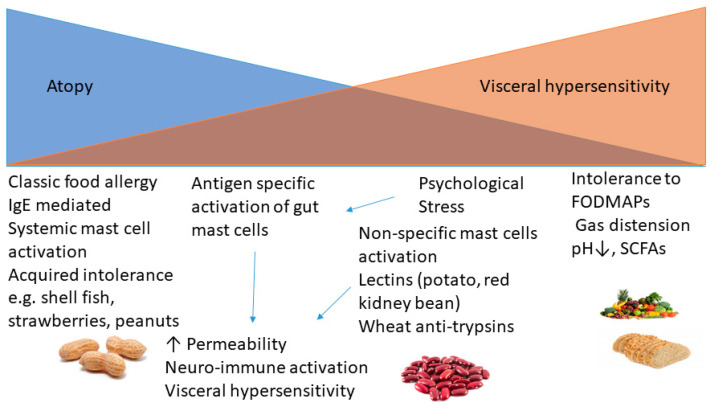
Immune and non-immune mechanisms underlying food intolerance in IBS.

**Figure 2 nutrients-13-00575-f002:**
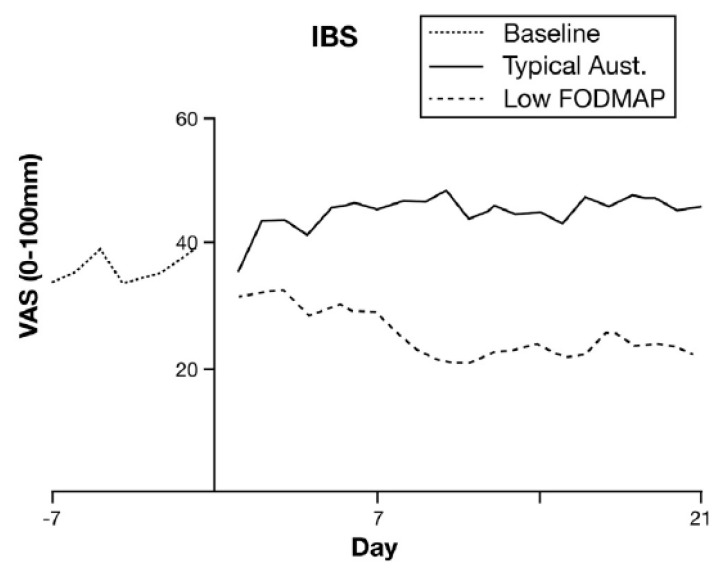
Mean overall gastrointestinal symptoms over time from start of either low FODMAP die or typical Australian diet high in FODMAPs. Symptom were significantly lower on the low FODMAP diet. Reproduced with permission from the Editor of Gut from Halmos et al. [92].

**Figure 3 nutrients-13-00575-f003:**
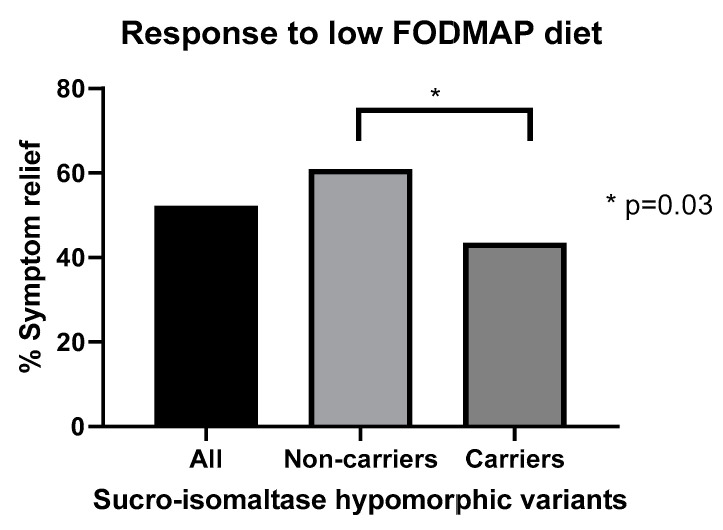
Impact of sucrose-isomaltase genetic variants which reduce enzyme activity. Reproduced with permission from Zheng et al. [76].

**Figure 4 nutrients-13-00575-f004:**
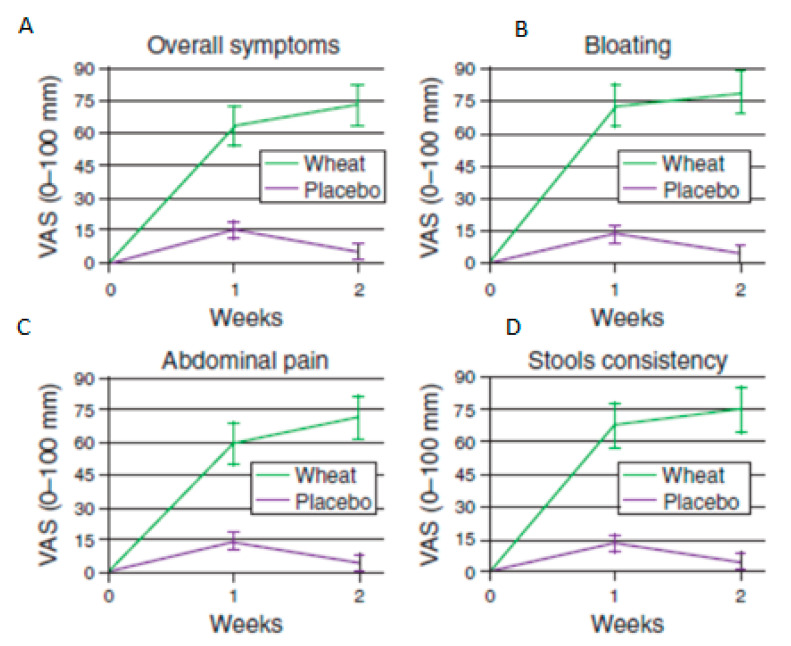
Response to double-blind, placebo-controlled wheat challenge in 270 wheat-sensitive non-celiac IBS patients from symmptom diary. *Y* axis shows symptom scores on a visual analog scale 0–100, where 0= no symptoms and 100 equals maximum symptoms for. (**A**) Overall symptoms, (**B**) Bloating, (**C**) Abdominal pain and (**D**) Stool consistency. Reproduced with permission of the editor from Carroccio et al. [114].

**Table 1 nutrients-13-00575-t001:** Commonly identified foods which caused symptoms [64].

Onions	35%
Milk	32%
Wheat	30%
Chocolate	28%
Butter	25%
Yoghurt	25%
Coffee	24%
Eggs	23%
Nuts	18%
Citrus	18%

## Data Availability

Not applicable.

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
