# Peer review of "Impact of Diet on Symptoms of the Irritable Bowel Syndrome"

_nutrients, 2021, doi:10.3390/nu13020575_

Round 1

Reviewer 1 Report

This is an excellent review. The text is very useful for a broad spectrum of stake holders, such as clinicians, medical/dietetic students, patient advocate groups and researchers. 

The paper is well written, up to date, comprising key clinical features of IBS and dietary measures. It also neatly combines clinical experiences to the research data on pathophysiology. Especially, I read with delight the sections on mast cells, which might be the main drivers of symptoms in IBS. 

Overall comments

The title of the article should better reflect the contents of the paper. Words such as “role of eating, diet, nutrition or food” should be included in the title to give a reader more realistic comprehension what to expect. The role of hypnotherapy, cognitive behavioral therapy, exercise and drug therapy are namely left out from the paper.

Would be interesting to know why the author opted for leaving the role of probiotics and peppermint oil capsules out of the paper. I think there is rather much data showing they are slightly better than placebo in IBS, and commonly recommended.

The paper describes the role of milk (esp. lactose) which is often thought to be one key driver of symptoms. However, the paper does not mention two RCTs showing hydrolyzed casein might be better tolerated than intact milk protein in IBS/other functional GI disorders. Casein rather than whey protein of milk seems to drive mast cell activation in vitro. Hydrolyzed casein also caused smaller histamine release than intact casein. Scand J Gastroenterol 1991 Apr;26(4):379-84. Another, very recent paper (https://doi.org/10.1002/mnfr.202000250) suggested that hydrolyzed casein accelerated upper GI transit in animals. In keeping with these observations, a RCT showed hydrolyzed casein reduced heartburn like symptoms, and overall GI symptoms mildly when compared to intact milk protein (DOI: 10.3390/nu12072140). The discussion on the role of casein in milk would be valuable addition.

There is a need for studies that scrutinize the potential effect of reducing carbon dioxide, caffeine, supplementing glutamine and/or fiber on top of low FODMAP diet. In clinical practice, many clinicians/dieticians seem to recommend combining all, or some, of these measures, but this approach of combining many treatment modalities has not been tested in adequate clinical trials. Perhaps, this could be mentioned in the conclusions.

It is debatable if the suggested treatment algorithm (“Conclusions”) is the one that should be followed in most cases. Sometimes, a simple test of glutamine or fiber supplementation, might be enough, or hypnotherapy. 

Role of kiwi fruits in IBS-C and lettuce, small bowel water, in IBS-D?

Further points

Rows 52-53. What is the time span, X days per ?(/weeks/months/year/lifetime)?

Rows 99-99. I would add that the potential role of lectins and agglutinins has never been addressed in relevant randomized studies. More data is needed, but is is good to be mentioned (to stimulate further studies).

Rows 171-187. Food can cause delayed symptoms. Some studies have shown that the symptoms of today may have their origin in yesterday. This makes it even more difficult for patients/clinicians to tease out what are the real trigger foods. Mego M, et al. Accumulative effect of food residues on intestinal gas production. Neurogastroenterol Motil. 2015;27:1621-8.

Row 212. I would specify wheat bran. As far as I know, oat bran and rye bran, have not been tested in  RCTs among IBS patients. Theoretically, oat bran/beta-glucan could be beneficial according to some recent animal data but humans trials are lacking. Also, this section would benefit from a notion that inulin, FOS and GOS also belong to soluble fibers but are more likely to cause symptoms than resolve them. Recommendation to simply use soluble fiber is thus outdated. 

Row 456. SERT -> first time, describe

Row 469->

Dutch cohort demonstrated that people with NCGS are 3.5-15x more likely to report poor tolerance to different high FODMAP foods. van Gils T et al. Prevalence and Characterization of Self-Reported Gluten Sensitivity in The Netherlands. Nutrients. 2016 Nov 8;8(11). How often NCGS is simply IBS –patients only want to call themselves wheat/gluten intolerant?  Is this more socially acceptable than IBS (sometimes has had a bad reputation)?

Row 514. Histamine free diets -> Low histamine diets

Author Response

Response to reviewer 1

Many thanks for you helpful and detailed comments which I have responded to below as follows

This is an excellent review. The text is very useful for a broad spectrum of stake holders, such as clinicians, medical/dietetic students, patient advocate groups and researchers. 

The paper is well written, up to date, comprising key clinical features of IBS and dietary measures. It also neatly combines clinical experiences to the research data on pathophysiology. Especially, I read with delight the sections on mast cells, which might be the main drivers of symptoms in IBS. 

Overall comments

The title of the article should better reflect the contents of the paper. Words such as “role of eating, diet, nutrition or food” should be included in the title to give a reader more realistic comprehension what to expect. The role of hypnotherapy, cognitive behavioral therapy, exercise and drug therapy are namely left out from the paper.

Response: I agree  The title is now modified to “Impact of diet on symptoms of the Irritable bowel syndrome”

Would be interesting to know why the author opted for leaving the role of probiotics and peppermint oil capsules out of the paper. I think there is rather much data showing they are slightly better than placebo in IBS, and commonly recommended.

Response: The article’s focus was deliberately about diet If it were to include therapy other than dietary it would be excessively long and exceed my allocated word limit

The paper describes the role of milk (esp. lactose) which is often thought to be one key driver of symptoms. However, the paper does not mention two RCTs showing hydrolyzed casein might be better tolerated than intact milk protein in IBS/other functional GI disorders. Casein rather than whey protein of milk seems to drive mast cell activation in vitro. Hydrolyzed casein also caused smaller histamine release than intact casein. Scand J Gastroenterol 1991 Apr;26(4):379-84. Another, very recent paper (https://doi.org/10.1002/mnfr.202000250) suggested that hydrolyzed casein accelerated upper GI transit in animals. In keeping with these observations, a RCT showed hydrolyzed casein reduced heartburn like symptoms, and overall GI symptoms mildly when compared to intact milk protein (DOI: 10.3390/nu12072140). The discussion on the role of casein in milk would be valuable addition.

Response: thanks for this  I had not included any comment about casein but it is certainly of interest even if not entirely clear what the mode of action is.  I now add a paragraph.as follows:  Milk intolerance is commoner than just lactose malabsorption and in children there is clear evidence of a true allergy to cows milk protein which is reduced by hydrolysing the casein component[81]. Classical IgE mediated milk protein allergy is rare in adults but recent trials have suggested that hydrolysing milk protein can reduce flatulence in patients with functional bowel disorders [82, 83]. It is unclear if this is due to gut specific allergy but this has been recently demonstrated in mice [20] and previously in coeliac patients, who responded to rectal instillation of milk protein despite lacking elevated  specific serum IgE[84]

There is a need for studies that scrutinize the potential effect of reducing carbon dioxide, caffeine, supplementing glutamine and/or fiber on top of low FODMAP diet. In clinical practice, many clinicians/dieticians seem to recommend combining all, or some, of these measures, but this approach of combining many treatment modalities has not been tested in adequate clinical trials. Perhaps, this could be mentioned in the conclusions.

Response:  I agree these measures are part of  the NICE and other  guidelines based on consensus rather than evidence. I have added this to the section on th e NICE diet “ It is worth noting that these measures are based on consensus rather than evidence since to test all possible variations in a randomised controlled way would require prohibitive numbers to achieve adequate statistical power. “

It is debatable if the suggested treatment algorithm (“Conclusions”) is the one that should be followed in most cases. Sometimes, a simple test of glutamine or fiber supplementation, might be enough, or hypnotherapy. 

Response: These conclusions are definitely a personal recommendation rather than evidenced based but I do think that readers appreciate some definite guidance awaiting definitive trial results which may be a long time coming.   I now emphasize this as follows in the conclusion changes underlined  : If this fails then there are many possible options. My personal approach is to start with a trial of a simple exclusion diet excluding just a few common items like wheat, milk, excessive caffeine and/ or specific foods the patient has identified which is a relatively easy first step

Role of kiwi fruits in IBS-C and lettuce, small bowel water, in IBS-D?

Response: Thank you for this comments I now include a new paragraph mentioning these studies

Laxative effects of fruit and vegetables

Recent imaging studies have shown that lettuce and rhubarb both markedly increase water content of the small bowel and the ascending colon [121]. This secretory response is likely to represent a response to specific compounds such as lactucins and rhein, produced by plants to inhibit grazing by herbivores. Those with IBS-D might benefit from avoiding these as do ileostomists[122]. By contrast those with constipation might benefit from such effects which are also seen with whole fruit such as apples [123]and kiwifruit[124]. This mechanistic evidence for kiwifruit is also supported by several randomised placebo controlled trials showing the increase in stool water is associated with increased bowel frequency and softening of stools[125-127]

Further points

Rows 52-53. What is the time span, X days per ?(/weeks/months/year/lifetime)?

Response: I now clarify this was a 3 month study and the average duration of  individual episodes was given in days . The text now reads.   “Symptoms typically occurs in episodes and in a 3 month study of 185 IBS patients average episode durations were 2.1 days for diarrhea, 4.5 days for constipation, 3.1 days for pain, and 3.5 days for bloating [10].”

Rows 99-99. I would add that the potential role of lectins and agglutinins has never been addressed in relevant randomized studies. More data is needed, but is is good to be mentioned (to stimulate further studies).

Response: Agreed  The text now reads “These gut mast cells can also be activated by psychological stressors[21] as well as directly by lectins such as raw potatoes[22] and agglutinins from uncooked red kidney beans[23] though these laboratory studies have yet to be shown to be clinically relevant by properly controlled trials of dietary intervention”

Rows 171-187. Food can cause delayed symptoms. Some studies have shown that the symptoms of today may have their origin in yesterday. This makes it even more difficult for patients/clinicians to tease out what are the real trigger foods. Mego M, et al. Accumulative effect of food residues on intestinal gas production. Neurogastroenterol Motil. 2015;27:1621-8.

Response: Thank you for this I have now included this valid  point  line 183-5 Furthermore adverse reactions to food such as excessive gas may be delayed[49] making it difficult for the patient to identify the food responsible

Row 212. I would specify wheat bran. As far as I know, oat bran and rye bran, have not been tested in  RCTs among IBS patients. Theoretically, oat bran/beta-glucan could be beneficial according to some recent animal data but humans trials are lacking. Also, this section would benefit from a notion that inulin, FOS and GOS also belong to soluble fibers but are more likely to cause symptoms than resolve them. Recommendation to simply use soluble fiber is thus outdated. 

Response: Yes I agree with these points and have altered the text  referring specifically to wheat bran and altering description of “soluble” fibre to more specifically “ viscous” fibre to avoid the confusion you so rightly point out. The text now reads from  line 216  “However two meta-analyses agree that this effect is only seen in trials of viscous fibre and particulate fibre like wheat bran shows no benefit [58, 59].  Recent studies show how viscous fibre (psyllium) acts by trapping water in the small bowel and increasing colonic water content leading to softer more frequent stools  [60].  Why viscous fibre would benefit constipation is obvious but why it also benefits non-constipated patients is unclear unless viscous fibre slows digestion and reduces rapid fermentation”.

Row 456. SERT -> first time, describe

Response: Done

Row 469->

Dutch cohort demonstrated that people with NCGS are 3.5-15x more likely to report poor tolerance to different high FODMAP foods. van Gils T et al. Prevalence and Characterization of Self-Reported Gluten Sensitivity in The Netherlands. Nutrients. 2016 Nov 8;8(11). How often NCGS is simply IBS –patients only want to call themselves wheat/gluten intolerant?  Is this more socially acceptable than IBS (sometimes has had a bad reputation)?

Response: Hopefully the text does indicate that there is an extensive overlap with IBS. What is needed is a noninvasive technique to objectively measure the response to wheat and or gluten

I now add this to the conclusions Line 563 Future research should focus on developing patient acceptable, non-invasive ways of reliably identifying specific food intolerances especially wheat and milk protein intolerance together with ways of reducing both immune and non-immune based adverse responses to food in IBS

Row 514. Histamine free diets -> Low histamine diets

Response: I agree you cannot get histamine free I have changed the title

Reviewer 2 Report

This is an interesting topic of growing interest due to the high prevalence of Irritable Bowel Syndrome (IBS). Although in recent years several excellent reviews covering a very similar scope have been published, this review is well written and organized and, especially, introduced explanations for contradicting results underlying the pathophysiology of IBS.

Few comments :

In the Histamine Free diet section, I suggest underlining the difficulty in achieving consensus on the diagnostic criteria for histamine intolerance. (Reese I et al German guideline for the management of adverse reactions to ingested histamine: Guideline of the German Society for Allergology and Clinical Immunology (DGAKI), the German Society for Pediatric Allergology and Environmental Medicine (GPA), the German Association of Allergologists (AeDA), and the Swiss Society for Allergology and Immunology (SGAI) Allergo J Int. 2017;26(2):72-79. doi: 10.1007/s40629-017-0011-5).

I agree with the need to explore the presence of ED. Therefore, due to the bidirectional relationship between EDs and GI disorders, where should physicians focus their attention to disclose an ED? 

Can a patient’s daily dairy help physicians to better detect false beliefs and abnormal eating behavior?

Author Response

Response to reviewer 2

Many thanks for your helpful comments

My response is below

This is an interesting topic of growing interest due to the high prevalence of Irritable Bowel Syndrome (IBS). Although in recent years several excellent reviews covering a very similar scope have been published, this review is well written and organized and, especially, introduced explanations for contradicting results underlying the pathophysiology of IBS.

Few comments :

In the Histamine Free diet section, I suggest underlining the difficulty in achieving consensus on the diagnostic criteria for histamine intolerance. (Reese I et al German guideline for the management of adverse reactions to ingested histamine: Guideline of the German Society for Allergology and Clinical Immunology (DGAKI), the German Society for Pediatric Allergology and Environmental Medicine (GPA), the German Association of Allergologists (AeDA), and the Swiss Society for Allergology and Immunology (SGAI) Allergo J Int. 2017;26(2):72-79. doi: 10.1007/s40629-017-0011-5).

Response:  Thank you I agree the literature is contradictory  and unsatisfactory.  I now add reference to these very helpful guidelines as follows

As recent guidelines indicate, despite much inaccurate comment in the social media, there are no objective measures to define histamine intolerance. Nor can the degradation pathways via diamino-oxidase or histamine-N methyl transferase be adequately measured at present. Furthermore there are no randomised trials of either diet nor pharmacological treatment so much work remains to be done before any firm recommendations can be made[123] 

I agree with the need to explore the presence of ED. Therefore, due to the bidirectional relationship between EDs and GI disorders, where should physicians focus their attention to disclose an ED? 

Can a patient’s daily dairy help physicians to better detect false beliefs and abnormal eating behavior?

Response: I agree a daily diary to be brought back to a subsequent consultation can be very helpful  I have added this line 195 If this is suspected then asking the patient to keep a daily food diary to bring to the next appointment can be very helpful by demonstrating a very restricted diet. 

Reviewer 3 Report

Current review entitled Irritable bowel syndrome is a fantastic manuscript  that summarize major research findings  about dietary triggers and IBS. 

The only major thing i would like to point is the Title of the work, which is not specific , as the whole review mainly focus on diet effects on IBS, so i kindly ask the author to change it to " Association between diet and Irritable Bowel syndrome" or something else.   Some minor points are the following:   - No data about the role of pro and prebiotics, especially from recent large scale trials ( Martoni et al. Nutrients 2020,  Lactobacillus acidophilus DDS-1 and Bifidobacterium lactis UABla-12 Improve Abdominal Pain Severity and Symptomology in Irritable Bowel Syndrome: Randomized Controlled Trial) - Please kindly add the work from  Laatikainen et al. Nutrients 2020 ( Randomised Controlled Trial: Partial Hydrolysation of Casein Protein in Milk Decreases Gastrointestinal Symptoms in Subjects with Functional Gastrointestinal Disorders) in the chapter about lactose restriction.(from line 291 - )   Please consider noting some things   about the work from  Bellini  et al. Nutrients 2020 (A Low-FODMAP Diet for Irritable Bowel Syndrome: Some Answers to the Doubts from a Long-Term Follow-Up) to the section of low fodmap diet) after Line 334
  - Please change the phrase Histamine free diets  to  Low histamine diets, which is more suitable (line 514)

Author Response

Response to reviewer 3

Many thanks for reading the manuscript and your helpful comments

My response  to your comments is below

Current review entitled Irritable bowel syndrome is a fantastic manuscript  that summarize major research findings  about dietary triggers and IBS. 

The only major thing i would like to point is the Title of the work, which is not specific , as the whole review mainly focus on diet effects on IBS, so i kindly ask the author to change it to " Association between diet and Irritable Bowel syndrome" or something else.  

Response: I agree  The title is now modified to “Impact of diet on symptoms of the Irritable bowel syndrome”

 Some minor points are the following:   - No data about the role of pro and prebiotics, especially from recent large scale trials ( Martoni et al. Nutrients 2020,  Lactobacillus acidophilus DDS-1 and Bifidobacterium lactis UABla-12 Improve Abdominal Pain Severity and Symptomology in Irritable Bowel Syndrome: Randomized Controlled Trial) –

Response: While agreeing that this is of interest the article’s focus was deliberately about diet If it were to include therapy other than dietary it would be excessively long and exceed my allocated word  limit

 Please kindly add the work from  Laatikainen et al. Nutrients 2020 ( Randomised Controlled Trial: Partial Hydrolysation of Casein Protein in Milk Decreases Gastrointestinal Symptoms in Subjects with Functional Gastrointestinal Disorders) in the chapter about lactose restriction.(from line 291 - )   -

Response: I had not included any comment about casein but it is certainly of interest even if it is not entirely clear what the mode of action is.  I now add a paragraph on line 324.as follows:  Milk intolerance is commoner than just lactose malabsorption and in children there is clear evidence of a true allergy to cows milk protein which is reduced by hydrolysing the casein component[81]. Classical IgE mediated milk protein allergy is rare in adults but recent trials have suggested that hydrolysing milk protein can reduce flatulence in patients with functional bowel disorders [82, 83]. It is unclear if this is due to gut specific allergy but this has been recently demonstrated in mice [20] and previously in coeliac patients, who responded to rectal instillation of milk protein despite lacking elevated  specific serum IgE[84]

 Please consider noting some things   about the work from  Bellini  et al. Nutrients 2020 (A Low-FODMAP Diet for Irritable Bowel Syndrome: Some Answers to the Doubts from a Long-Term Follow-Up) to the section of low fodmap diet) after Line 334

Response: Thanks for alerting me to this useful new paper.  I now reference it on line 382

It is also important to recognise that patients should not remain on the strict exclusion diet but once symptoms have responded they should reintroduce suspect foods for a challenge period and thus enable a liberalizing of diet. Few studies have dealt with this phase of the treatment but a recent report showed  that around 50% of patients were successful in achieving symptom relief.  The study showed the importance of a dietician because many of the initial perceived intolerances were not confirmed during systematic reintroduction[95].  
  - Please change the phrase Histamine free diets  to  Low histamine diets, which is more suitable (line 514)

Response: I agree you cannot get histamine free I have changed the title